# Role of Genetic Polymorphism Present in Macrophage Activation Syndrome Pathway in Post Mortem Biopsies of Patients with COVID-19

**DOI:** 10.3390/v14081699

**Published:** 2022-07-31

**Authors:** Aline Cristina Zanchettin, Leonardo Vinicius Barbosa, Anderson Azevedo Dutra, Daniele Margarita Marani Prá, Marcos Roberto Curcio Pereira, Rebecca Benicio Stocco, Ana Paula Camargo Martins, Caroline Busatta Vaz de Paula, Seigo Nagashima, Lucia de Noronha, Cleber Machado-Souza

**Affiliations:** 1Faculdades Pequeno Príncipe, Av. Iguaçu, 333, Curitiba 80230-020, Paraná, Brazil; aline.zanchettin78@gmail.com (A.C.Z.); leovinicius@live.com (L.V.B.); 2Instituto de Pesquisa Pelé Pequeno Príncipe, Av. Silva Jardim, 1632, Curitiba 80250-200, Paraná, Brazil; 3School of Medicine, Pontifícia Universidade Católica do Paraná, R. Imaculada Conceição, 1155, Curitiba 80215-901, Paraná, Brazil; andersonazevedodutra@yahoo.com.br (A.A.D.); danimargarita@yahoo.com.br (D.M.M.P.); marcos.curcio@pucpr.edu.br (M.R.C.P.); rebecca.stocco@pucpr.edu.br (R.B.S.); anapaulacamargo@hotmail.com (A.P.C.M.); carolbvaz@gmail.com (C.B.V.d.P.); seigo_nagashima@hotmail.com (S.N.); lnno.noronha@gmail.com (L.d.N.)

**Keywords:** SARS-CoV-2, secondary hemophagocytic lymphohistiocytosis, macrophage, immunohistochemistry, polymorphisms

## Abstract

COVID-19 is a viral disease associated with an intense inflammatory response. Macrophage Activation Syndrome (MAS), the complication present in secondary hemophagocytic lymphohistiocytosis (sHLH), shares many clinical aspects observed in COVID-19 patients, and investigating the cytolytic function of the responsible cells for the first line of the immune response is important. Formalin-fixed paraffin-embedded lung tissue samples obtained by post mortem necropsy were accessed for three groups (COVID-19, H1N1, and CONTROL). Polymorphisms in MAS cytolytic pathway (*PRF1*; *STX11*; *STXBP2*; *UNC13D* and *GZMB*) were selected and genotyping by TaqMan^®^ assays (Thermo Fisher Scientific, MA, USA) using Real-Time PCR (Applied Biosystems, MA USA). Moreover, immunohistochemistry staining was performed with a monoclonal antibody against perforin, CD8+ and CD57+ proteins. Histopathological analysis showed high perforin tissue expression in the COVID-19 group; CD8+ was high in the H1N1 group and CD57+ in the CONTROL group. An association could be observed in two genes related to the cytolytic pathway (*PRF1* rs885822 G/A and *STXBP2* rs2303115 G/A). Furthermore, *PRF1* rs350947132 was associated with increased immune tissue expression for perforin in the COVID-19 group. The genotype approach could help identify patients that are more susceptible, and for this reason, our results showed that perforin and SNPs in the *PRF1* gene can be involved in this critical pathway in the context of COVID-19.

## 1. Introduction

In 2020, a new type of infection, caused by the *Coronaviridae* family virus (severe acute respiratory syndrome coronavirus 2—SARS-CoV-2), proved to be highly contagious and caused many hospitalizations and deaths, especially in people with comorbidities and older age. Host-related risk factors have been identified as associated with developing the severe form of COVID-19 (Coronavirus Disease 19) [1].

The laboratory alterations of COVID-19 included lymphopenia, hypoalbuminemia, alterations in the lactate dehydrogenase enzyme, C-reactive protein, ferritin, and D-dimer. In these cases, an exacerbation of the inflammatory process is observed, defined as a “cytokine storm”. The hypercytokinemia process resembles the complication called Macrophagic Activation Syndrome (MAS), also described in children with the severe form of COVID-19. Hemophagocytic lymphohistiocytosis (HLH) is characterized by a fulminant cytokine storm leading to multiple organ dysfunction and high mortality. HLH is classified into familial (fHLH) and into secondary (sHLH). Secondary, or acquired, hemophagocytic lymphohistiocytosis is present in some rheumatologic diseases (Juvenile Idiopathic Arthritis—JIA, Systemic Lupus Erythematosus—SLE and Still’s Disease), and the MAS may occur similar to a complication in sHLH. The MAS is often used by rheumatologists to describe a potentially serious complication of the immune system that causes inflammatory disease or hypercytokinemia [2]. Consequently, the SARS-CoV-2 infection produces clinical symptoms that resemble those observed in rheumatologic conditions associated with MAS/sHLH.

MAS is characterized by an imbalance in the cytotoxic function, reducing the cytolytic function involved with Natural Killer (NK) cells and cytotoxic T lymphocytes (CD8+). There would be a proliferation of T lymphocytes (CD4+ and CD8+) that would contribute to the expressive increase of pro-inflammatory cytokines, which reinforces the presence of hypercytokinemia. Furthermore, there would be excessive activation of inflammatory cells at later times, especially with macrophages [3]. Interestingly, these changes lead to characteristic clinical that are very similar to those found in patients with the severe form of COVID-19.

On the other hand, the familial or primary form of hemophagocytic lymphohistiocytosis (fHLH/pHLH) occurs due to autosomal recessive defects in genes (*PRF1*; *UNC13D*; *STX11*; *STXBP2*) encoding proteins involved in cytotoxic granule exocytosis of NK-induced apoptosis [4,5,6]. Regarding the sHLH, patients may present the fundamental genetic polymorphisms described for pHLH [7]. Thus, this article hypothesizes that gene polymorphisms, present in the context of MAS/sHLH, may be associated in patients with the severe form of COVID-19. Analysis of these polymorphisms in COVID-19 patients could help identify the window of opportunity for premature immunosuppressive therapies similar to those used in treating HLH.

## 2. Materials and Methods

### 2.1. Study Population

The present study was approved by the National Research Ethics Committee (Conselho Nacional de Ética em Pesquisa—CONEP 3.944.734/2020 for COVID-19 patients and 2.550.445/2018 for H1N1 and CONTROL patients). The authors confirm that all methods were carried out following relevant guidelines and regulations. Families permitted the post-mortem biopsy of the cases of COVID-19, H1N1, and CONTROL groups; and signed the informed consent forms. The sample collection followed all relevant ethics and safety protocols.

The pandemic COVID-19 group (n = 24) comprises lung samples from post-mortem biopsies of patients whose cause of death was SARS-CoV-2 diffuse alveolar damage during the 2020 outbreak in the ICU at Hospital Marcelino Champagnat in Curitiba-Brazil. Clinical details about this sample can be accessed in papers in the group [8,9,10].

Lung samples from *post mortem* biopsies from patients whose cause of death was H1N1pdm09 (Pandemic disease caused by Influenza A Virus, H1N1 subtype) severe acute respiratory disease during the 2009 outbreak in the Intensive Care Unit (ICU) at Hospital de Clínicas in Curitiba-Brazil, constitute the H1N1 group (n = 10) positive control. Testing for H1N1 and SARS-CoV-2 was performed on nasopharyngeal swabs taken during ICU hospitalization, and real-time reverse transcriptase-polymerase chain reaction (RT-PCR) was positive in all cases.

A CONTROL group (n = 10), negative control, was composed of lung samples from necropsies of patients who died due to other causes (cardiovascular disease and cancer), not involving lung lesions in the same hospital above.

### 2.2. Genetic Analysis

The DNA was obtained from paraffinized cuts of the samples using a commercially available paraffin DNA extraction kit (Qiagen^®^_,_ Hilden, Germany). After determining the concentration, the samples will be diluted to a final concentration of 20 ng/µL for working solution and stored in a freezer at −20 °C with restricted access and only allowed to researchers involved in the project the technical personnel for them authorized.

The eight polymorphisms in the proposed genes were chosen following the authors’ strategy. First, the authors searched the qualified literature for articles that focused on SNP and MAS [11,12]. After that, an SNP target search tool (SNP info) was used that uses gene coverage concepts using linkage disequilibrium calculations [13]. After this search, the third moment was to observe whether the SNPs separated by the authors by reading the qualified articles were the same after using the SNP info. Four genes are associated with MAS, and three are associated with membrane receptors in COVID-19. After this search eight polymorphisms were selected: *PRF1* (perforin 1—rs10999426; rs885821; rs885822; rs35947132); *STX11* (syntaxin 11—rs7764017); *STXBP2* (syntaxin binding protein 2—rs6791; rs2303115); *UNC13D* (unc-13 homolog D—rs3744007) and *GZMB* (granzyme B—rs6573910). The patients’ purified DNA was amplified by real-time PCR (Applied Biosystems 7500 Real-Time PCR System; Thermo Fisher Scientific, MA USA). The TaqMan^®^ system of allelic discrimination is an essay in which genomic variants are detected through a multiplex polymerase chain reaction (PCR), which combines the amplification and detection of the polymorphic segment in a single step using probe oligonucleotides marked with different fluorescent chemistry (usually VIC^TM^ and FAM^TM^).

### 2.3. Immunohistochemistry Analysis

The lung samples provided by *post-mortem* biopsy were formalin-fixed paraffin-embedded (FFPE) and stained with hematoxylin and eosin (H&E). The immunohistochemistry technique was used to identify the expression of the perforin (primary antibodies for anti-Perforin—mouse monoclonal; 1:200 dilution; clone 5B10; BioSB^®^, Santa Barbara, CA USA), cytotoxic T lymphocyte (primary antibodies for anti-CD8—mouse rabbit; 1:100 dilution; clone SP16 cod MAS-14548; Thermo Fisher^®^, Thermo Fisher Scientific, MA USA) and natural killer—NK (primary antibodies for anti-CD57—mouse monoclonal; ready-to-use; clone TB01; Dako^®^, Santa Clara, CA USA). The immunohistochemical assay included both negative control (histological section of a hyperplasic lymph node, which the primary antibody was omitted) and positive control (hyperplasic lymph node). Tissue samples were incubated in primary antibodies in a humid chamber temperature between 2 and 8ºC, overnight. The secondary polymer (Dako Advance™ HRP System, DakoCytomation, Inc., CA, USA) was applied to the material tested for 30 min at room temperature. The technique was revealed by adding the 2, 3, diamino-benzidine complex + hydrogen peroxide substrate, for a brown color turning time, then, the counterstaining with Harris Hematoxylin was performed.

The anti-perforin immunostained slides were observed exclusively in the alveolar septum and perivascular spaces by counting lymphocytes cells in 20 randomized high-power field—HPF (40×, Olympus Objective, 0.26 mm^2^ per sample), through a BX50 optical microscope (OLYMPUS, Tokyo, Japan). Average scores were obtained by screening 20 randomized HPFs. The same methods were used for staining anti-CD8 and CD57 antibodies.

### 2.4. Statistical Analysis

The nominal variables were expressed by frequency/percentages and the non-nominal variables by means and standard deviation. The normality condition of the variables in each group was evaluated using the Shapiro-Wilks test, and the comparison of the quantitative variables of the two groups was performed using the non-parametric Kruskal Wallis test or U de Mann-Whitney test. The Student’s *t*-test was used to compare the results obtained in two qualitative variables groups. Values of *p* < 0.05 indicated statistical significance. Bonferroni correction was used for multiple independent genetic testing (models addictive dominant and recessive), and adjusted *p* values < 0.002 were considered significant only for genotype analysis. Spearman correlation (r) analysis was made in all three groups. Data were analyzed using the computer program by IBM^®^ SPSS Statistics v.20.0 software (Armonk, NY, USA: IBM Corp).

## 3. Results

There was a higher frequency of males in all three groups in this sample. Age and time from hospitalization to death in the COVID-19 group were higher than H1N1 and CONTROL groups (Table 1; Figure 1). In the COVID-19 group, the mean and standard deviation of lymphocytes was 1044.32 ± 817.41 (mg/dL). The tissue immunoexpression of perforin was higher (*p* = 0.001) in the COVID-19 group (3.91 ± 3.42) compared to the H1N1 group (1.02 ± 0.52). The IHQ expression for CD8+ was smaller in COVID-19 (19.9 ± 13.8) compared with the H1N1 (38.3 ± 24.5) groups (Table 1; Figure 2). The same aspect can be observed for CD57+ expression in COVID-19 (1.3 ± 1.1) compared with H1N1 (2.8 ± 1.3) groups (Table 1; Figure 2).

Correlations between the perforin and two cell types (cytotoxic T lymphocytes and NK) were performed individually for three groups (COVID-19, H1N1, and CONTROL). A moderate and positive significant correlation (r = 0.572; *p* = 0.003) was observed in the COVID-19 group when the immunohistochemical expression between perforin and CD8+ was analyzed. A similar result with a moderate and positive significant correlation (r = 0.442; *p* = 0.031) was observed between perforin and CD57+.

The distribution of gene frequencies for all tag SNPs can be observed for the addictive model in Table 2 and for dominant and recessive models in Table 3. *PRF1* rs885822 G/A increases heterozygous genotype (GA) in COVID-19 vs. CONTROL groups (Table 2). The *STXBP2* rs2303115 G/A showed the GG genotype was more frequent in COVID-19 vs. the H1N1 groups. 

Regarding the correlation of tissue immunoexpression of perforin with the four polymorphisms in the *PRF1* gene, it can be observed that the highest tissue expression values were associated with the GA genotype (rs35947132) in the COVID-19 group (Table 4).

## 4. Discussion

The current accumulated knowledge in COVID-19 physiopathogenic pathways showed that the damage to the lower respiratory tract caused by SARS-CoV-2 was, in most cases, followed by the presence of hyper inflammation [14]. Moreover, the elevated levels of inflammatory markers were correlated with unfavorable outcomes such as diffuse alveolar damage and mortality [15].

Several studies have suggested a physiopathogenic role of the monocytes and macrophages in COVID-19 and hypercytokinemia [16], but not yet fully understood [17]. A recent review suggested that COVID-19-associated hyperinflammatory syndrome may have significant pathogenic overlap with virus-induced MAS/sHLH. This aspect is associated with macrophage activation with high production of cytokines and involvement of NK and CD8+ T cells [14].

The association of COVID-19 with HLH, due to the similarity of some inflammatory aspects, could early identify the patient most likely to develop this severe form. Gene regulation, which leads to inter-individual differences in the basal levels of the immune response, presents a diversity of markers with the capacity to produce different immunophenotypes. Some of these markers are strongly associated with an increased risk of immune responses and thus provide important information about pathological mechanisms [18,19,20]. Our results should be interpreted with caution, but they seem to be promising as they point to possible biomarkers associated with the context of COVID-19. In this preliminary study, polymorphisms in genes described in secondary MAS/HLH showed biological plausibility in our analyses. 

The clinical and demographic characteristics of the patients with COVID-19 and H1N1pdm09, such as the incubation time of the disease, length of hospital stay, viral clearance, and treatments instituted can be observed in Table 1. Regarding tissue expression of perforin (Table 1), there was a significant difference when comparing COVID-19 with H1N1 (*p* = 0.000) and CONTROL (*p* = 0.000) groups. Since the tissue expression of perforin does not represent the evolution of the disease, but a photograph of the moment of death of the patients, these differences could be explained by some hypotheses. However, when we observe the CONTROL group, we have a “constitutive” expression value of cytoplasmic and membranous perforin (Figure 1), which is higher than the expression values of the H1N1 group, which may reflect the degradation process of this protein after its use to induce apoptosis during H1N1pdm09 infection [21]. Although, this reasoning does not apply to the COVID-19 group since these perforin values should also be lower in infected patients due to the described degradation process. In addition, patients with severe forms of COVID-19 often have lymphopenia associated with lower CD8+ and NK cell counts and, consequently, it should have lower perforin expression, supported by the theory of immunological exhaustion in persistent infections [22]. The literature also indicates the opposite of exhaustion, which would be a process of hyperactivation of CD8+ T lymphocytes [23]. Alterations involving cytotoxic mechanisms, such as the movement of vesicles containing perforin and granzyme, could justify the observation of higher perforin values in the COVID-19 group [21]. These alterations, involving cytotoxic mechanisms, could be directly associated with genetic defects in molecules involved in the cytolytic process. The genotyping findings of this study could try explaining the higher tissue expression values of perforin in the COVID-19 group. Polymorphisms in the perforin gene could produce dysfunctional proteins, leading to a blunting of the apoptosis process and, consequently, greater perforin expression as a compensation mechanism.

The *PRF1* rs885822 (G/A) were found associated with susceptibility to multiple sclerosis [24], survival in childhood acute lymphoblastic leukemia [25], and HIV-1 vertical transmission [26]. This SNP has a benign functional consequence [27] because it is a synonymous/missense variant [28] and presents a global minor allele frequency (GMAF) of 0.3041 for the wild G allele. In our results, the G allele frequency was 0.4062, which can be reflected in heterozygous frequency [28]. Other studies showed that the presence of heterozygous can modify the perforin stability and function [29,30] and recently Cabrera-Merrante and colleagues (2020) related two patients with COVID-19 with the *PRF1* Ala91Val polymorphism (rs35947132), showing the importance in this gene [31]. Furthermore, this specific classical *PRF1* Ala91Val polymorphism (rs35947132) was related to HLH risk [32]. This SNP (rs35947132 G/A) in our study (Table 2) shows that there may be an enrichment in the heterozygote (GA) frequency of the COVID-19 group. It is also worth noting that there was no heterozygote of this SNP in the H1N1 and CONTROL groups (Table 2). These observed results could be important in the COVID-19 clinical context. In our study, the rs885822 (G/A) showed the highest frequency of heterozygosity (GA) in the COVID-19 group (Table 2) compared to the CONTROL group (*p* = 0.029), which could be responsible for a perforin dysfunctional expression.

Another gene associated with MAS/sHLH was *STXBP2*. The intron variant rs2303115 (G/A) showed the highest frequency of heterozygosity (GA) in the COVID-19 group compared to the H1N1 group (*p* = 0.007) (Table 2). The rs2303115 presenting a global minor allele frequency (GMAF) for the wild G allele was 0.6438. In our results, the G allele frequency was 0.6666 [33]. This SNP has nonclinical significance reported [33]. This gene (*STXBP2*) encodes a protein called syntaxin biden protein 2 involved in intracellular trafficking to release cytotoxic granules by natural killer cells. Mutations in this gene are associated with pHLH [34,35,36] and other conditions like myocardial infarction [37]. These significant results observed in the SNPs of *PRF1* and *STXBP2* genes lose statistical significance after applying the Bonferroni correction. However, its biological plausibility leads us to consider these two important results still in scope.

Table 4 shows the correlation between perforin tissue expression and the SNP of the *PRF1* gene. It can be observed that the highest values of perforin tissue expression were found in the COVID-19 group, specifically in the heterozygous (GA) genotype in the rs350947132 (7.78 ± 2.84). Another interesting result is that two of the four SNPs addressed were associated with higher protein immunoexpression in the heterozygous genotype. This aspect is important because the loss of heterozygosity (LOH) can lead to the clinical manifestations of the disease phenotype, similar to what has been described in studies related to neoplasms. An example of this effect can be seen in the work of van de Vijver and Nusse in 1991, in which the authors compare DNA from breast carcinomas with DNA from normal cells from the same patient and detect the loss of heterozygosity for several loci [38]. The literature shows that about 30% of patients with pHLH, with a deficiency in cytotoxic function, have a specific mutation in the gene that encodes perforin [39]. Positive correlations were observed in the COVID-19 group between the perforin and two cell types (cytotoxic T lymphocytes and NK), and these results confirm the cytotoxic cellular function [40].

This preliminary study has some limitations. The first is the small sample size for genetic analysis in association studies, but it is important to note that this sample was recruited from post-mortem biopsies at critical moments in both pandemics (COVID-19 and H1N1). Still considering the limitation of the sample N, the authors used the Bonferroni Correction (BC) specifically for genetic analyses. This statistical tool minimizes the possibility of a type 1 error. For this reason, all significant *p*-values were lost. However, our results show biological plausibility when analyzed before applying the Bonferroni correction. Immunohistochemistry associated with SNPs provides only a momentary picture of the outcome, and thus functional studies could further help elucidate the physiopathogenesis associated with COVID-19. Another limitation could be the age difference between the groups, but unfortunately, this aspect is the reality of our sample.

The discovery of the possible genetic biomarkers could enable their use for the early identification (window of opportunity) of individuals most susceptible to the worst outcome. In addition, the genetic biomarkers could help identify candidate patients for early treatment. Our results could help identify patients with clinical overlapping between COVID-19 and MAS/sHLH since the literature already recommends a triple-differentiated therapeutic (corticosteroids; cyclosporine A; etoposide) approach recently added to the use of biological agents (anti-IL1B; anti-IL-6; anti-IFNγ) for severe COVID-19 patients with MAS clinical manifestations [38,39,41,42].

## Figures and Tables

**Figure 1 viruses-14-01699-f001:**
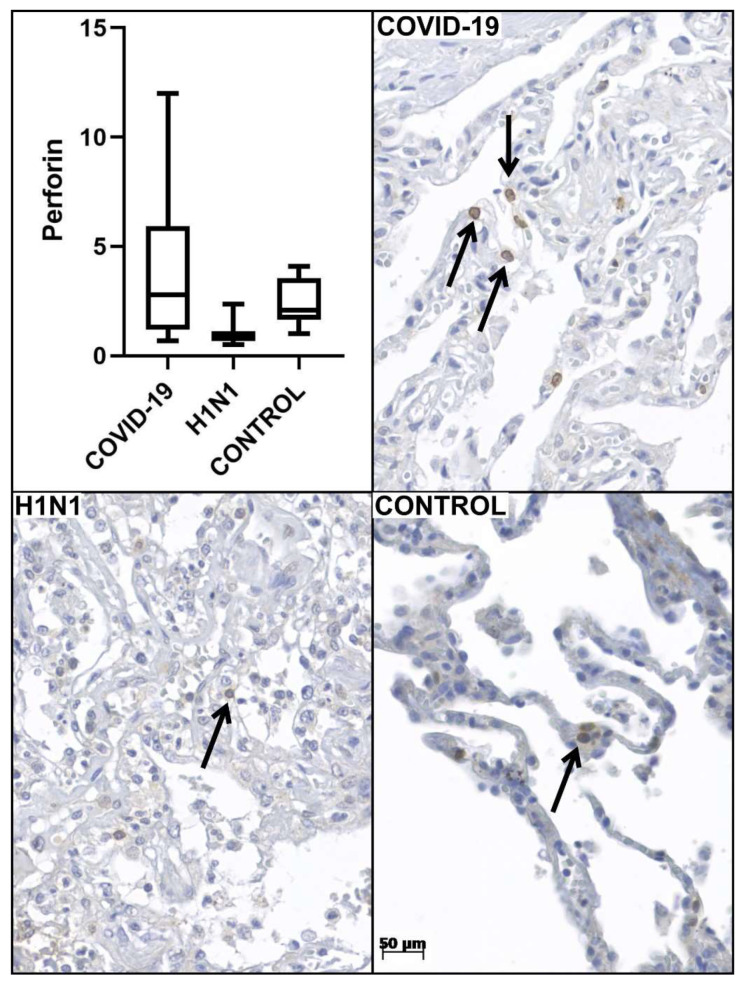
Graphics are showing tissue immunoexpression of perforin (number of perforin + cells per high-power fields—HPF) for COVID-19, H1N1, and CONTROL groups. Photomicrography shows T lymphocytes (**arrows**) expressing perforin in all three groups (40× HPF).

**Figure 2 viruses-14-01699-f002:**
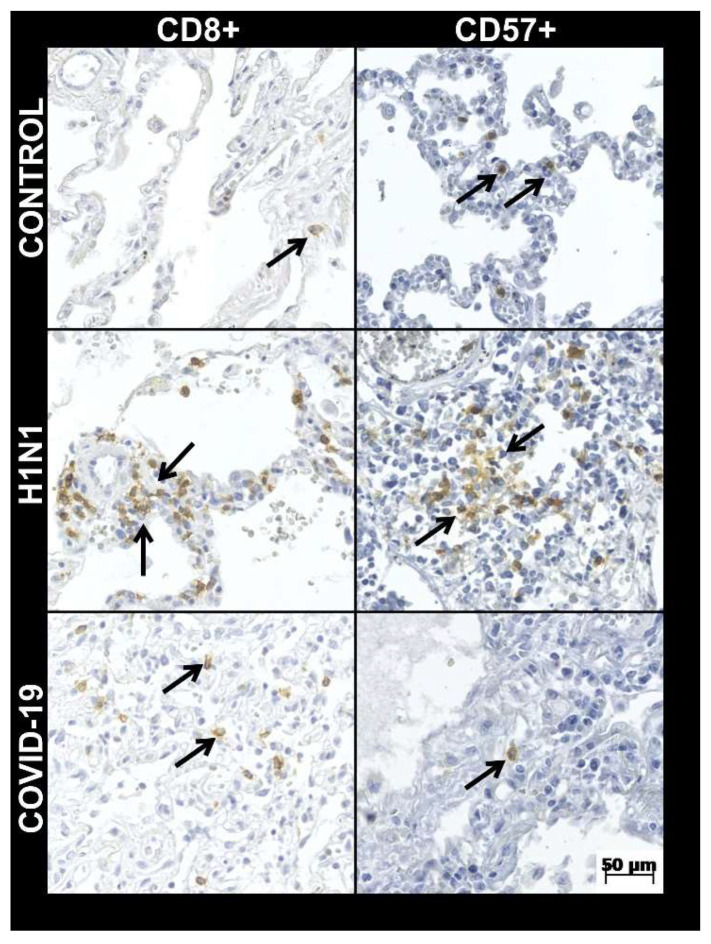
Tissue immunoexpression of CD8+ and CD57+ (number of positive cells per high-power fields—HPF) for COVID-19, H1N1, and CONTROL groups in 40× HPF (arrows show immunostained cells).

**Table 1 viruses-14-01699-t001:** Baseline characteristics in the study population.

Variables	COVID-19 (n = 24)	H1N1 (n = 10)	*p*-Value ^1^	Control (n = 10)	*p*-Value ^2^
**Age ***	70.7 ± 13.0	41.7 ± 16.0	0.000 ^a^	44.7 ± 12.4	0.000 ^a^
**Gender **** Male	13 (45.2)	8 (80.0)	0.675 ^b^	7 (70.0)	1.000 ^b^
Female	11 (45.8)	2 (20.0)		3 (30.0)	
**Time from hospitalization to death (days) ***	15.2 ± 10.4	4.7 ± 6.1	0.001 ^a^	3.8 ± 3.5	0.000 ^a^
**Perforin tissue expression ***	3.9 ± 3.4	1.0 ± 0.5	0.000 ^a^	2.8 ± 1.0	0.001 ^a^
**CD8+ tissue expression ***	19.9 ± 13.8	38.3 ± 24.5	0.000 ^a^	10.7 ± 4.6	0.000 ^a^
**CD57+ tissue expression ***	1.3 ± 1.1	2.8 ± 1.3	0.000 ^a^	4.7 ± 2.9	0.012 ^a^

CD8+: Cytotoxic T Lymphocyte; CD57+: Natural Killer; * Mean ± Standard Deviation; ** Absolute number (percentage); ^1^ COVID-19 vs. H1N1 groups; ^2^ COVID-19 vs. CONTROL groups; ^a^ Mann-Whitney U; ^b^ Fisher’s Exact Test.

**Table 2 viruses-14-01699-t002:** Genotypic analysis between 3 groups (COVID-19, H1N1, and CONTROL) for *PRF1, STX11, STXBP2, UNC13D,* and *GZMB* genes in the addictive model.

Gene—Reference SNP ^†^ Allele Variation [1/2]	Homozygous 1/1	Heterozygous 1/2	Homozygous 2/2	*p*-Value */**
** *PRF1—* ** **rs10999426 [G/A]**	GG	GA	AA	
COVID-19	8 (33.3)	15 (62.5)	1 (4.2)	
H1N1	6 (60.0)	4 (40.0)	0 (0.0)	0.321 *
CONTROL	2 (50.0)	1 (25.0)	1 (25.0)	0.203 **
** *PRF1—* ** **rs885821 [G/A]**	GG	GA	AA	
COVID-19	17 (70.8)	5 (20.8)	2 (8.3)	
H1N1	6 (60.0)	4 (40.0)	0 (0.0)	0.380 *
CONTROL	7 (87.5)	1 (12.5)	0 (0.0)	0.574 **
** *PRF1—rs* ** **885822 [G/A]**	GG	GA	AA	
COVID-19	3 (12.5)	15 (62.5)	6 (25.0)	
H1N1	0 (0.0)	4 (40.0)	6 (60.0)	0.114 *
CONTROL	0 (0.0)	1 (16.7)	5 (83.3)	0.029 **
** *PRF1—* ** **rs35947132 [G/A]**	GG	GA	AA	
COVID-19	20 (83.3)	4 (16.7)	0 (0.0)	
H1N1	10 (100.0)	0 (0.0)	0 (0.0)	0.169 *
CONTROL	9 (100.0)	0 (0.0)	0 (0.0)	0.191 **
** *STX11* ** **—rs7764017 [A/G]**	AA	AG	GG	
COVID-19	11 (45.8)	11 (45.8)	2 (8.3)	
H1N1	5 (50.0)	5 (50.0)	0 (0.0)	0.642 *
CONTROL	1 (100.0)	0 (0.0)	0 (0.0)	0.569 **
** *STXBP2—* ** **rs6791 [A/G]**	AA	AG	GG	
COVID-19	4 (16.7)	12 (50.0)	8 (33.3)	
H1N1	0 (0.0)	3 (30.0)	7 (70.0)	0.108 *
CONTROL	1 (16.7)	1 (16.7)	4 (66.7)	0.277 **
** *STXBP2—* ** **rs2303115 [G/A]**	GG	GA	AA	
COVID-19	11 (45.8)	10 (41.7)	3 (12.5)	
H1N1	2 (22.2)	1 (11.1)	6 (66.7)	0.007 *
CONTROL	0 (0.0)	0 (0.0)	1 (100.0)	0.569 **
** *UNC13D—* ** **rs3744007 [G/A]**	GG	GA	AA	
COVID-19	0 (0.0)	1 (4.3)	22 (95.7)	
H1N1	0 (0.0)	0 (0.0)	10 (100.0)	1.000 *
CONTROL	0 (0.0)	0 (0.0)	10 (100.0)	1.000 **
** *GZMB* ** **—rs6573910 [C/T]**	CC	CT	TT	
COVID-19	5 (20.8)	13 (54,2)	6 (25.0)	
H1N1	1 (10.0)	6 (60.0)	3 (30.0)	0.749 *
CONTROL	2 (66.7)	0 (0.0)	1 (33.3)	0.145 **

**^†^** SNP identifier based on NCBI dbSNP; Genotype was expressed by number and percentage and a total percentage was shown in line; * COVID-19 vs. H1N1 groups; ** COVID-19 vs. CONTROL groups; Logistic regression *p*-value. Values of *p* < 0.05 indicated statistical significance, but after Bonferroni correction, the *p*-value < 0.002 can be considered significant. The *p*-value before the Bonferroni correction is underlined.

**Table 3 viruses-14-01699-t003:** Genotypic distribution for *PRF1, STX11, STXBP2* and *GZMB* genes in dominant and recessive models.

Gene Reference SNP ^†^ Allele Variation	Models		COVID-19 (n = 24)	H1N1 (n = 10)	*p*-Value *	CONTROL (n = 10)	*p*-Value **
** *PRF1* **	Dom G	GG + GA	23 (95.8)	10 (100.0)	0.512 ^b^	3 (75.0)	0.134 ^b^
**rs10999426**		AA	1 (4.2)	0 (0.0)		1 (25.0)	
**G/A**	Rec G	AA + GA	16 (66.7)	4 (40.0)	0.150 ^b^	2 (50.0)	0.520 ^b^
		GG	8 (33.3)	6 (60.0)		2 (50.0)	
** *PRF1* **	Dom G	GG + GA	22 (91.7)	10 (100.0)	0.347 ^b^	8 (100.0)	0.399 ^b^
**rs885821**		AA	2 (8.3)	0 (0.0)		0 (0.0)	
**G/A**	Rec G	AA + GA	7 (29.2)	4 (40.0)	0.538 ^b^	1 (12.5)	0.346 ^b^
		GG	17 (70.8)	6 (60.0)		7 (87.5)	
** *PRF1* **	Dom G	GG + GA	18 (75.0)	4 (40.0)	0.112 ^b^	1 (16.7)	0.016 ^b^
**rs885822**		AA	6 (25.0)	6 (60.0)		5 (83.3)	
**G/A**	Rec G	AA + GA	21 (87.5)	10 (100.0)	0.242 ^b^	6 (100.0)	0.361 ^b^
		GG	3 (12.5)	0 (0.0)		0 (0.0)	
** *STX11* **	Dom A	AA + AG	22 (91.7)	10 (100.0)	0.347 ^b^	1 (100.0)	0.763 ^b^
**rs7764017**		GG	2 (8.3)	0 (0.0)		0 (0.0)	
**A/G**	Rec A	GG + AG	13 (54.2)	5 (50.0)	0.824 ^a^	0 (0.0)	0.288 ^b^
		AA	11 (45.8)	5 (50.0)		1 (100.0)	
** *STXBP2* **	Dom A	AA + AG	16 (66.7)	3 (30.0)	0.068 ^b^	2 (33.3)	0.136 ^b^
**rs6791**		GG	8 (33.3)	7 (70.0)		4 (66.7)	
**A/G**	Rec A	GG + AG	20 (83.3)	10 (100.0)	0.169 ^b^	5 (83.3)	1.000 ^b^
		AA	4 (16.7)	0 (0.0)		1 (16.7)	
** *STXBP2* **	Dom G	GG + GA	21 (87.5)	3 (33.3)	0.005 ^b^	1 (100.0)	0.706 ^b^
**rs2303115**		AA	3 (12.5)	6 (66.7)		0 (0.0)	
**G/A**	Rec G	AA + GA	13 (54.2)	7 (77.8)	0.216 ^b^	0 (0.0)	0.288 ^b^
		GG	11 (45.8)	2 (22.2)		1 (100.0)	
** *GZMB* **	Dom C	CC + CT	18 (75.0)	7 (70.0)	0.763 ^b^	2 (66.7)	0.756 ^b^
**rs6573910**		TT	6 (25.5)	3 (30.0)		1 (33.3)	
**C/T**	Rec C	TT + TC	19 (79.2)	9 (90.0)	0.450 ^b^	1 (33.3)	0.156 ^b^
		CC	5 (20.8)	1 (10.0)		2 (66.7)	

**^†^** SNP identifier based on NCBI dbSNP; Dom, Dominant model; Rec, Recessive model; Genotype was expressed by number and percentage and a total percentage was shown in column; * COVID-19 vs. H1N1 groups *p*-value; ** COVID-19 vs. CONTROL groups *p*-value; ^a^ Pearson Chi-Square. ^b^ Fisher’s Exact Test. Values of *p* < 0.05 indicated statistical significance, but after Bonferroni correction, the *p*-value <0.002 can be considered significant. The *p*-value before the Bonferroni correction is underlined.

**Table 4 viruses-14-01699-t004:** Correlation between perforin tissue expression and genotyping in *PRF1* gene in COVID-19 and H1N1 group.

Reference SNP ^†^ and Allele Variation [1/2]	Homozygous 1/1	Heterozygous 1/2	Homozygous 2/2
**rs10999426 [G/A]**	GG	GA	AA
COVID-19 *	1.26 ± 0.70	4.72 ± 3.58	NA
H1N1 *	1.20 ± 0.62	0.75 ± 0.14	NA
**rs885821 [G/A]**	GG	GA	AA
COVID-19 *	4.06 ± 3.69	2.24 ± 1.24	0.75 ± 0.28
H1N1*	1.06 ± 0.65	0.96 ± 0.35	NA
**rs885822 [G/A]**	GG	GA	AA
COVID-19 *	4.40 ± 6.27	4.06 ± 3.09	1.28 ± 0.83
H1N1 *	NA	0.75 ± 0.14	1.20 ± 0.62
**rs35947132 [G/A]**	GG	GA	AA
COVID-19 *	2.53 ± 2.68	7.78 ± 2.84	NA
H1N1 *	1.02 ± 0.52	NA	NA

**^†^** SNP identifier based on NCBI dbSNP; * Mean ± Standard Deviation for perforin tissue expression; NA: not available.

## Data Availability

Not applicable.

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
