# Peer review of "Role of Genetic Polymorphism Present in Macrophage Activation Syndrome Pathway in Post Mortem Biopsies of Patients with COVID-19"

_viruses, 2022, doi:10.3390/v14081699_

Round 1

Reviewer 1 Report

Role of genetic polymorphism present in macrophage activation syndrome pathway in post mortem biopsies of patients with COVID-19

Aline C. Zanchettin 1-2, Leonardo V. Barbosa 1-2, Anderson A. Dutra 3, Daniele M. M. Prá 3, Marcos R. C. Pereira 3, Rebecca B. Stocco 3, Ana Paula Camargo Martins 3, Caroline B. Vaz de Paula 3, Seigo Nakashima 3, Lucia de Noronha 3, Cleber Machado-Souza

General Comments:

This paper by Zanchettin et al attempts to explain the phenomenon of hypercytokinemia in patients with COVID 19 by studying genetic polymorphisms and tissue expression in post mortem lung biopsies of patients with COVID19, H1N1 and control patients. The authors found that there was high tissue expression of perforin in post mortem lung biopsies in patients with COVID 19, and this was associated with particular polymorphisms of PRF1 gene. While the results of this study are certainly intriguing there are limitations to the conclusions in this study:

-       The clinical significance of perforin expression is unclear – is this seen as well in other patients with MAS/sHLH, or is this seen in patients with paucisymptomatic or mild COVID infection – inclusion of these control groups would strengthen the paper

-       The COVID 19 group is significantly older – age matched controls of lung specimens without COVID/H1N1 would be interesting

The study conclusion (identification or particular polymorphisms in PRF1, STXBP1) could be  strengthened by inclusion of those control groups. Functional correlates of perforin expression could be done in peripheral blood as well, if available and that way non-post mortem samples of paucicymptomatic infected patients could also be studied. 

Overall while this study highlights an interesting finding, the significance of the results are unclear due to low sample size, lack of statistical significance and omission of control groups that would help validate the finding of PRF tissue expression and correlated genetic polymorphism as an important clinical biomarker. 

Specific Comments:

Introduction:

·      Line 42 -  “MAS can be associated with systemic autoimmune diseases such as secondary or acquired hemophagocytic lymphohistiocytosis (sHLH)” – This statement is confusing, and it may be more accurate to say “associated with systemic autoimmune disease such as systemic JIA etc”

·      Line 61 – “Regarding the sHLH, patients may present the fundamental genetic alterations described for pHLH”- Does this mean to say genetic polymorphisms

·      Spelling check “hypercitokinemia”- ?hypercytokinemia

·      This section is unclear, syntax could be improved 

Materials and Methods:

-       N for all groups relatively low – why was this sample size chosen? Was any power calculation done

-       Line 96 – requires a reference

-       Line 99 – search strategy and SNP selection method should be included in appendix data, what data was included to settle on these variants

Results:

-       Table 1 – Baseline characteristics – more clinical information about the subject is required – what were the diagnoses of the non-infected post mortem controls, could this affect perforin/CD8/CD57 expression?

- additionally what was the pulmonary disease and systemic disease burden  of the infected patients - did they all have ARDS? did they have sHLH?

-       Other clinical characteristic of patients that could be included are total lymphocyte count as this could also affect tissue immunohistochemistry results

-       As mentioned above inclusion of different control groups would be interesting (HLH or MAS not secondary to COVID 19)  

-       Tissue sample from COVID 19 subjects are significant older – is there any age-related difference in perforin/CD8+/CD57+ expression?

-       Tables 2/3 – Unclear why certain p values are bolded and underlined – footnotes state that p < 0.002 is significant

-       Table 4 – are these correlations statistically significant?

Discussion

-       Line 226 – are there any references that study tissue expression of perforin in the context of HLH? Those would be helpful in determining the significance of results

-       Line 228-229 - The clinical and demographic characteristics of the patients with 228 COVID-19 and H1N1pdm09, such as the incubation time of the disease, length of hospital stay, viral clearance and treatments instituted. – This sentence seems incomplete 

-       Line 243 – What genetic defects are associated with different perforin expression

-       Line 290-292 – Unfortunately this is big limitation of this study, and many of the results do not receive statistical significance (P< 0.002 is stated in the table footnotes). If this has been misinterpreted and the results do have statistical significance, that should be made clearer

Author Response

Dear Editor of Viruses

We thank the Reviewers for the precious suggestions and the opportunity to improve our manuscript ID viruses-1786252 entitled "Role of genetic polymorphism present in macrophage activation syndrome pathway in post mortem biopsies of patients with COVID-19".

The authors made the changes in a separate manuscript (uploaded as "viruses-1786252_revised manuscript.docx") and highlighted all changes in yellow. Below we forward point-by-point the following answers to your questions:

Reviewer: 1

This paper by Zanchettin et al. attempts to explain the phenomenon of hypercytokinemia in patients with COVID 19 by studying genetic polymorphisms and tissue expression in post-mortem lung biopsies of patients with COVID19, H1N1 and control patients. The authors found that there was high tissue expression of perforin in post-mortem lung biopsies in patients with COVID 19, which was associated with polymorphisms of the PRF1 gene. While the results of this study are undoubtedly intriguing, there are limitations to the conclusions in this study:

-     The clinical significance of perforin expression is unclear – is this seen as well in other patients with MAS/sHLH, or is this seen in patients with paucisymptomatic or mild COVID infection – inclusion of these control groups would strengthen the paper

Authors: Perforin is an essential protein in the cytolytic function considering the immune response, and PRK1 (perforin gene) is mutated in rheumatological diseases (LHHp and LHHs) that produce inflammatory conditions. For this reason, the authors chosen had to focus on this protein. This study aimed to evaluate the tissue expression of perforin in the primary target organ of COVID-19, that is, the lung. The samples in this study are post-mortem lung samples of fatal COVID-19. The surgical procedure to obtain lung samples in patients with mild/moderate COVID19 is not indicated due to invasive procedure, with no clinical indication and ethically debatable. For this reason, the inclusion of this other group was not possible, as suggested by the reviewer.

The COVID 19 group is significantly older – age-matched controls of lung specimens without COVID/H1N1 would be interesting.

Authors: The samples in this study were collected during the first wave of COVID-19 in Brazil (April-August 2020), with older patients constituting most deaths at that time. Lung samples from patients with Influenza A virus, H1N1 subtype, were collected in June-August 2009 during the H1N1pdm09 pandemic. The demographic characteristics of patients who died from severe pneumonia caused by H1N1pdm09 were utterly different from those observed in patients who died from severe COVID-19, with the H1N1pdm09 pandemic affecting mainly young patients, making it impossible to pair these two groups. To compose the control group, we used a mixture of patients with cardiovascular and neoplastic causes of death (with lung samples without lesions) available in our necropsy sample tissue bank. However, within our necropsy bank, we have rare older adults (over 70 years old, comparable to patients in the COVID-19 group), and all of the few elderly patients available had bacterial pneumonia as the cause of death making its use as a control impracticable. Therefore, our control group represents our best choice of available samples. The authors included this difference in the limitation paragraph (Line 303-304).  

The study conclusion (identification or polymorphisms in PRF1, STXBP1) could be strengthened by including those control groups. Functional correlates of perforin expression could be done in peripheral blood as well, if available, and that way, non-post mortem samples of paucisymptomatic infected patients could also be studied.

Authors: The authors are grateful for the reviewers' comments and agree that questions about the other methods could be interesting to improve the results and conclusions. However, this study aimed to evaluate the tissue expression of perforin in the primary target organ of COVID-19, that is, the lung.

Overall while this study highlights an interesting finding, the significance of the results is unclear due to the low sample size, lack of statistical significance and omission of control groups that would help validate the finding of PRF tissue expression and correlated genetic polymorphism as an important clinical biomarker.

Authors: The authors thank the reviewer's comments. Aware of the low sample size, the authors used terms that only suggest the result's relevance.

Specific Comments:

Introduction:

Line 42 -  “MAS can be associated with systemic autoimmune diseases such as secondary or acquired hemophagocytic lymphohistiocytosis (sHLH)” – This statement is confusing, and it may be more accurate to say “associated with systemic autoimmune disease such as systemic JIA etc."

Authors: The authors agree with the reviewer and change this paragraph.

Line 61 – “Regarding the sHLH, patients may present the fundamental genetic alterations described for pHLH”- Does this mean to say genetic polymorphisms

Authors: The authors changed the word as suggested by the reviewer.

Spelling check “hypercitokinemia”- ?hypercytokinemia

Authors: The authors corrected the term throughout all text.

This section is unclear; the syntax could be improved.

Authors: The authors wrote again and tried to improve their English.

Materials and Methods:

N for all groups relatively low – why was this sample size chosen? Was any power calculation done?

Authors:  The post-mortem samples were accessed by minimally invasive necropsy performed in patients that died due to the severe form of the disease during the COVID-19 and H1N1 pandemic. These are lung tissue samples from a particular group of patients, which includes the difficulties in obtaining them, such as the family's permission to perform the necropsy, the performance of the necropsy in an ICU environment soon after death, the risks of team contagion, among others. Therefore, these samples are precious in their characteristics, although we know they are few. However, this aspect was reported as a limitation at the end of the discussion. The authors did not do the power calculation.

Line 96 – requires a reference.

Authors: The authors insert the reference as suggested by the reviewer.    

Line 99 – search strategy and SNP selection method should be included in appendix data; what data was included to settle these variants.

Authors: The authors would like to maintain these data in methods because it is essential to understand the selection context.

Results:

Table 1 – Baseline characteristics – more clinical information about the subject is required – what were the diagnoses of the non-infected post-mortem controls? Could this affect perforin/CD8/CD57 expression?

Authors: The authors write about the CONTROL group in Line 91 (die for cardiovascular disease and cancer). The authors used the uninfected control group to obtain a baseline value of the studied markers. However, the authors are aware that these lung samples, although not showing tissue damage on anatomopathological examination, belong to hospitalized patients who died in the hospital. There is no doubt that these facts could modify the expression of the studied biomarkers. However, these control group patients died from cardiovascular or neoplastic causes and did not present with chronic or acute lung injury at the time of autopsy. Given the infeasibility of obtaining lung samples from healthy patients for obvious reasons, this was the best control we could use.

Additionally, what was the pulmonary and systemic disease burden of the infected patients - did they all have ARDS? Did they have sHLH?

Authors: In the Methods session, the authors reported that both patients in the COVID-19 and H1N1 groups died from diffuse alveolar damage due to infection (Line 80), therefore, evolved with acute respiratory distress syndrome (ARDS). Regarding the second question, patients in both COVID-19 and H1N1 groups did not have the necessary score to be diagnosed with sHLH.

Another clinical characteristic of patients that could be included is total lymphocyte count, which could also affect tissue immunohistochemistry results.

Authors: The authors are grateful for the reviewer's observation and add the Mean ± Standard Deviation for lymphocyte count value in the results section (Line 151).

As mentioned above inclusion of different control groups would be interesting (HLH or MAS not secondary to COVID 19)  

Authors: The authors are grateful for the reviewer's observation.    

Tissue samples from COVID 19 subjects are significantly older – is there any age-related difference in perforin/CD8+/CD57+ expression?

Authors: Advanced age is a common condition regarding the severity of respiratory viral diseases, and this severity could be associated with altered T cell responses. Cellular senescence may also be associated with age, contributing to ineffective responses to a viral infection. As T cells replicate numerous times to the detriment of pathogen stimulation during the whole life of the host, they may lose the ability to proliferate and reach the stage of replicative senescence, according to the loss of telomerase activity (Xu W, Larbi A. Markers of T Cell Senescence in Humans. Int J Mol Sci. 2017 Aug 10;18(8):1742. doi: 10.3390/ijms18081742.)

Tables 2/3 – Unclear why certain p values are bolded and underlined – footnotes state that p < 0.002 is significant

Authors: The authors agreed and included a statistical analysis section (Line 142) that 0.05 is the p-value. It was included in the footnote in tables 2 and 3 that the p-value underlined is for the p-value before the Bonferroni correction. The authors remove the bolded p-value for tables 2 and 3.    

Table 4 – are these correlations statistically significant?

Authors: In table 4, the authors only wanted to indicate the values ​​of perforin expression in the possible genotypes. Furthermore, most cells in Table 4 had "NA" (NA: not available), making the p-value analysis entirely inappropriate.

Discussion

Line 226 – are there any references that study tissue expression of perforin in the context of HLH? Those would be helpful in determining the significance of the results.

Authors: the authors can’t find this reference. The only expression found is by RNAm or flow cytometry.

Line 229 - The clinical and demographic characteristics of the patients with COVID-19 and H1N1pdm09, such as the incubation time of the disease, length of hospital stay, viral clearance and treatments instituted. – This sentence seems incomplete 

Authors: The reviewer is correct. This sentence is incomplete. The authors made the modifications as observed by the reviewer and moved to Line 231. ·    

Line 243 – What genetic defects are associated with different perforin expression

Authors: The authors change the phrase (Line 250) to "These alterations, involving cytotoxic mechanisms, could be directly associated with genetic defects in molecules involved in the cytolytic process." ·    

Line 290-292 – Unfortunately, this is a big limitation of this study, and many of the results do not receive statistical significance (P< 0.002 is stated in the table footnotes). If this has been misinterpreted and the results do have statistical significance, that should be made clearer.

Authors: The authors tried to rewrite this important aspect observed by the reviewer. Bonferroni correction for genetics was used to guarantee the fairness of our findings. The authors could not use this correction to make the statistical results interesting but not accurate. But the authors reinforce at many points throughout the text that this is a preliminary study but with exciting developments from biological plausibility.

Reviewer 2 Report

Reviewer #1: 

Coronavirus disease 2019 (COVID-19) has emerged as a new world pandemic, infecting millions of people with a substantial mortality. There is significant interest study to analyzed the role of genetic polymorphism present in macrophage activation syndrome pathway.

Recently publications show several SNPs related to clinical severity of the patient.

In this manuscript, by Aline C. Zanchettin et al titled “Role of genetic polymorphism present in macrophage activation syndrome pathway in post mortem biopsies of patients with COVID-19”

The authors performed an analysis of genetic polymorphism present in macrophage activation syndrome pathway finding two genes related to cytolytic pathway (PRF1 and STXBP2), and just perforin and SNPs in PRF1 gene can be involved in this critical pathway 26 in the context of COVID-19.

There are several concerns that to be addressed.

This manuscript is well written and sites key findings in the field, therefore it will be helpful for clinical investigators entering into coronavirus/COVID-19 research. The study would benefit the section on information on the genetic mechanisms of critical illness in COVID-19.

Comments to improve the clarity of the manuscript are provided below.

Comments for the authors' consideration:

·      Could be interesting increase the N in the COVID-10 and H1N1 groups.

·      In the figure 1 and 2 could be used other cell markers.

·      In other studies, include inflammation genes could be interest.

·      Levels of some interleukins could be analyzed.

Author Response

Reviewer: 2

Coronavirus disease 2019 (COVID-19) has emerged as a new world pandemic, infecting millions of people with substantial mortality. There is a significant interest study to analyze the role of genetic polymorphism present in the macrophage activation syndrome pathway.

Recently publications show several SNPs related to the clinical severity of the patient.

This manuscript by Aline C. Zanchettin et al. titled "Role of genetic polymorphism present in macrophage activation syndrome pathway in post mortem biopsies of patients with COVID-19."

The authors performed an analysis of genetic polymorphism present in the macrophage activation syndrome pathway finding two genes related to the cytolytic pathway (PRF1 and STXBP2) and just perforin and SNPs in the PRF1 gene can be involved in this critical pathway 26 in the context of COVID-19.

There are several concerns that to be addressed.

This manuscript is well written and sites key findings in the field, therefore it will be helpful for clinical investigators entering into coronavirus/COVID-19 research. The study would benefit the section on information on the genetic mechanisms of critical illness in COVID-19.

Comments to improve the clarity of the manuscript are provided below.

Comments for the authors' consideration:

Could be interesting to increase the N in the COVID-10 and H1N1 groups.

Authors: We would like to acknowledge the pertinent considerations provided by the reviewer. This is a preliminary study with few samples. Still, the authors inform that they are trying to partner with other studies that address post-mortem biopsies in COVID19 and H1N1, but unfortunately, still without concrete success. We have very clearly the importance of this article since the relationship between COVID-19 and SAM is a topic addressed by eminent researchers.

In figures 1 and 2 could be used other cell markers.

Authors: The authors agree with the reviewer and report that we are currently seeking financial support to improve the analysis of other important markers and molecules associated with the process.

In other studies, including inflammation genes could be interesting.

Authors: The authors agree with the reviewer.

Levels of some interleukins could be analyzed.

Authors: The authors agree with the reviewer.

Round 2

Reviewer 1 Report

Thank you for addressing the comments from the previous round. Most of the previous comments have been adequately addressed, and if not addressed were listed as a limitation of the study. The only remaining unclear portion is in the methodology - line 105-105. It is understandable how these genes were chosen but it is unclear to the reader how these 8 polymorphisms were settled on. Inclusion of that rationale would be helpful. 

Author Response

The authors made the changes in the separate manuscript ("viruses-1786252_revised manuscript_V2.docx") and highlighted all changes in green. Below we forward the response to reviewer 1:

Reviewer: 1

Comments and Suggestions for Authors

Thank you for addressing the comments from the previous round. Most of the previous comments have been adequately addressed, and if not addressed were listed as a limitation of the study. The only remaining unclear portion is in the methodology - line 105-105. It is understandable how these genes were chosen but it is unclear to the reader how these 8 polymorphisms were settled on. Inclusion of that rationale would be helpful.

AUTHORS: The authors thank the reviewer for the above considerations and realize that the article is better than the initial version. The authors change the paragraph (Line 99).

The eight polymorphisms in the proposed genes were chosen following the authors' strategy. First, the authors searched the qualified literature for articles that focused on SNP and MAS [11,12]. After that, a SNP target search tool (SNP info) was used that uses gene coverage concepts using linkage disequilibrium calculations [13]. After this search, the third moment was to observe whether the SNPs separated by the authors by reading the qualified articles were the same after using the SNP info.
